# NRF2 loss recapitulates heritable impacts of paternal cigarette smoke exposure

**Patrick J. Murphy**[1,2☯]*, **Jingtao Guo**[2,3☯], **Timothy G. Jenkins**[3], **Emma R. James**[3,4], **John R. Hoidal**[5], **Thomas Huecksteadt**[5], **Dallin S. Broberg**[3], **James M. Hotaling**[3], **David F. Alonso**[6], **Douglas T. Carrell**[3,4,7], **Bradley R. Cairns**[2]*, **Kenneth I. Aston**[3]*

**1** Department of Biomedical Genetics, Wilmot Cancer Institute, University of Rochester Medical Center, Rochester, New York, United States of America, **2** Howard Hughes Medical Institute, Department of Oncological Sciences and Huntsman Cancer Institute, University of Utah School of Medicine, Salt Lake City, Utah, United States of America, **3** Andrology and IVF Laboratories, Department of Surgery, University of Utah School of Medicine, Salt Lake City, Utah, United States of America, **4** Department of Obstetrics and Gynecology, University of Utah School of Medicine, Salt Lake City, Utah, United States of America, **5** Department of Internal Medicine, University of Utah School of Medicine and Salt Lake VA Medical Center, Salt Lake City, Utah, United States of America, **6** Department of Psychology, University of Utah, Salt Lake City, Utah, United States of America, **7** Department of Genetics, University of Utah School of Medicine, Salt Lake City, Utah, United States of America

☯ These authors contributed equally to this work.

\* patrick_murphy@urmc.rochester.edu (PJM); brad.cairns@hci.utah.edu (BRC); kiaston@utah.edu (KIA)

**Data Availability Statement:** Genomics data is available through the NIH GEO Datasets under accession number GSE133742.

## Abstract

Paternal cigarette smoke (CS) exposure is associated with increased risk of behavioral disorders and cancer in offspring, but the mechanism has not been identified. Here we use mouse models to investigate mechanisms and impacts of paternal CS exposure. We demonstrate that CS exposure induces sperm DNAme changes that are partially corrected within 28 days of removal from CS exposure. Additionally, paternal smoking is associated with changes in prefrontal cortex DNAme and gene expression patterns in offspring. Remarkably, the epigenetic and transcriptional effects of CS exposure that we observed in wild type mice are partially recapitulated in *Nrf2*⁻/⁻ mice and their offspring, independent of smoking status. *Nrf2* is a central regulator of antioxidant gene transcription, and mice lacking *Nrf2* consequently display elevated oxidative stress, suggesting that oxidative stress may underlie CS-induced heritable epigenetic changes. Importantly, paternal sperm DNAme changes do not overlap with DNAme changes measured in offspring prefrontal cortex, indicating that the observed DNAme changes in sperm are not directly inherited. Additionally, the changes in sperm DNAme associated with CS exposure were not observed in sperm of unexposed offspring, suggesting the effects are likely not maintained across multiple generations.

## Author summary

Paternal cigarette smoking has been linked to increased risk of several behavioral disorders, obesity, and cancer in offspring, but the mechanisms by which this occurs have not been identified. *How does paternal cigarette smoke exposure, originating in the lungs, lead*

**Funding:** Funds for this research came from R01HD082062 through the NICHD, from HHMI, and from NCI CA042014 (for core facilities). The funders had no role in study design, data collection and analysis, decision to publish, or preparation of the manuscript.

**Competing interests:** The authors have declared that no competing interests exist.

*to systemic heritable health problems in offspring*? Here we address this question through genome-wide epigenetic and gene expression studies of mouse. We find that paternal cigarette smoke exposure has a substantial impact on offspring, including significant changes in gene expression and DNA methylation in the brains of unexposed F1 mice. We hypothesized that the observed heritable impacts were likely caused by systemic oxidative stress. Indeed, when we assessed epigenetic and gene expression patterns in mice that were null for the *Nrf2* gene (an established mouse model that is hyper-sensitized to oxidative stress) many of the observed outcomes from paternal cigarette smoke exposure were recapitulated, including impacts on epigenetic marks and gene expression in unexposed *Nrf2*$^{+/-}$ offspring. These results suggest that oxidative stress is an important mechanism by which the heritable negative impacts of paternal cigarette smoking arise.

## Introduction

Cigarette smoke (CS) exposure is a global epidemic with significant health consequences. In a recent study it was estimated that more than one third of the world's population is regularly exposed to tobacco smoke [1]. The health consequences of smoke exposure are significant and include numerous diseases and dysfunctions of the respiratory tract, increased risk of multiple types of cancer and cardiovascular disease [2, 3]. Tobacco smoke exposure is a global problem, the implications of which are becoming increasingly apparent. However, little is known about the impact of paternal exposure to CS on sperm and implications of preconception paternal CS on offspring health [4].

Male fertility rates have steadily declined in developed countries over the past half-century [5–7]. These trends are due to a variety of factors, but increased exposures to environmental toxins and negative lifestyle factors likely contribute [8, 9]. Cigarette smoking is associated with an accumulation of cadmium and lead in seminal plasma, reduced sperm count and motility, and increased morphological abnormalities in sperm [8, 9]. In addition, reduced reproductive potential has been reported in tobacco smoke-exposed mice [10] and humans [11]. Adult male mice exposed to sidestream tobacco smoke display significant increases in sperm DNA mutations at expanded simple tandem repeats (ESTRs) [12], as well as more frequent aberrations in sperm chromatin structure and elevated sperm DNA damage [10]. In contrast, CS-exposed male mice exhibit no measurable increase in somatic cell chromosome damage, indicating that germ cells may be more prone to environmentally-induced genetic and/or epigenetic insults compared with somatic cells.

The International Association for Research on Cancer has declared that *paternal* smoking *prior* to pregnancy is associated with a significantly elevated risk of leukemia in the offspring [13], suggesting CS-induced changes occur in sperm that influence offspring phenotype. Smoking has been clearly shown to modify DNAme patterns and gene expression in somatic tissues in individuals exposed to first- or second-hand tobacco smoke [14–16] as well as in newborns of smoking mothers [17–19]. We recently reported altered sperm DNA methylation (DNAme) patterns in men who smoke [20].

Mounting evidence supports sperm epigenetic changes as a mechanism for increased health risks in offspring of smoking fathers. We therefore aimed to explore the dynamics of sperm epigenetic changes after withdrawal from smoke, and to explore the mechanism underlying CS-induced alterations in offspring.

## Results

### Experimental design and phenotypic effects

Male mice were assigned to CS- exposed or non-exposed groups (n = 10–12 per group). The CS animals were exposed to the body mass-adjusted equivalent of 10–20 cigarettes per day, 5 days per week over a period of 60 days- corresponding to two complete cycles of spermatogenesis (Fig 1A). CS- exposed and control mice were bred to unexposed females, and offspring were analyzed for phenotypic and molecular measures. The nuclear factor (erythroid-derived 2)-like 2 (NRF2) pathway is the primary cellular defense against the cytotoxic effects of oxidative stress. Thus, to investigate the mode by which CS-induced epigenetic changes occur, we utilized the $Nrf2^{-/-}$ mouse model, which has compromised antioxidant capacity. In agreement with the literature, both wild type (WT) and $Nrf2^{-/-}$ mice that were exposed to CS weighed significantly less than non-exposed control animals (S1A Fig). Sperm concentration and motility were not significantly impacted by CS exposure, but sperm concentration was lower in $Nrf2^{-/-}$ than WT males, and conception was significantly delayed in CS-exposed $Nrf2^{-/-}$ mice compared with unexposed $Nrf2^{-/-}$ mice (S1A Fig). Growth trajectories, sperm parameters, litter size and sex ratios in offspring were not different in F1 animals based on paternal CS exposure or paternal genotype (S1B Fig).

### Smoking-induced DNAme changes in F0 sperm recovered in accordance with CG density

We performed reduced representation bisulfite sequencing (RRBS) to explore the effects of smoking on F0 sperm DNAme (sperm collected within 3 days of completing a 60-day smoking treatment), and we examined whether CS-induced DNAme changes recover to baseline, unexposed levels following removal of CS exposure (28, 103 and 171 days after smoking treatment; n = 10 per group). Due to the epigenetic reprogramming that occurs in early embryos, it is unlikely that CS-induced changes in paternal sperm DNAme directly influences offspring phenotype. However, because DNAme is a sensitive marker and can be reliably assessed on a genome-wide scale, we view it as an important signature of epigenetic change. Recent studies have found that increased variation in DNAme can be a consequence of particular environmental insults, including cigarette smoke exposure [21–24]. We therefore analyzed our data in a manner that would allow us to assess both average DNAme changes and variation in DNAme levels. Here, we first assessed DNAme at individual CpGs, and then investigated regional DNAme. We found CS-associated changes in DNAme at a large number of individual CpGs (S1E Fig) with essentially equal representation of sites that lost DNAme and gained DNAme. In addition, we found that the number of differentially methylated CpGs declined only slightly after removal of CS exposure for 28–171 days (S1C Fig). We partitioned differentially methylated CpGs into three groups: 1) shared, meaning differentially methylated CpGs that were maintained between the treatment group and the respective recovery group, 2) recovered, meaning CpGs that were differentially methylated after initial exposure but were no longer differentially methylated in the recovery group, and 3) new, meaning CpGs that were not differentially methylated after initial exposure but emerged as newly differentially methylated in the recovery groups. These classes were assessed separately for CpGs that initially lost DNAme verses those that initially gained DNAme relative to the control. For CpGs that lost DNAme, the three groups were relatively evenly represented across recovery times, with only a slight over-representation of shared CpGs. For CpGs that gained DNAme, there was a slight bias toward the emergence of new differentially methylated CpGs across recovery groups (S2D Fig).

Given that DNAme status of CpG regions likely has a greater capacity to confer functional effects compared with individual sites, we subsequently binned individual CpGs into regions

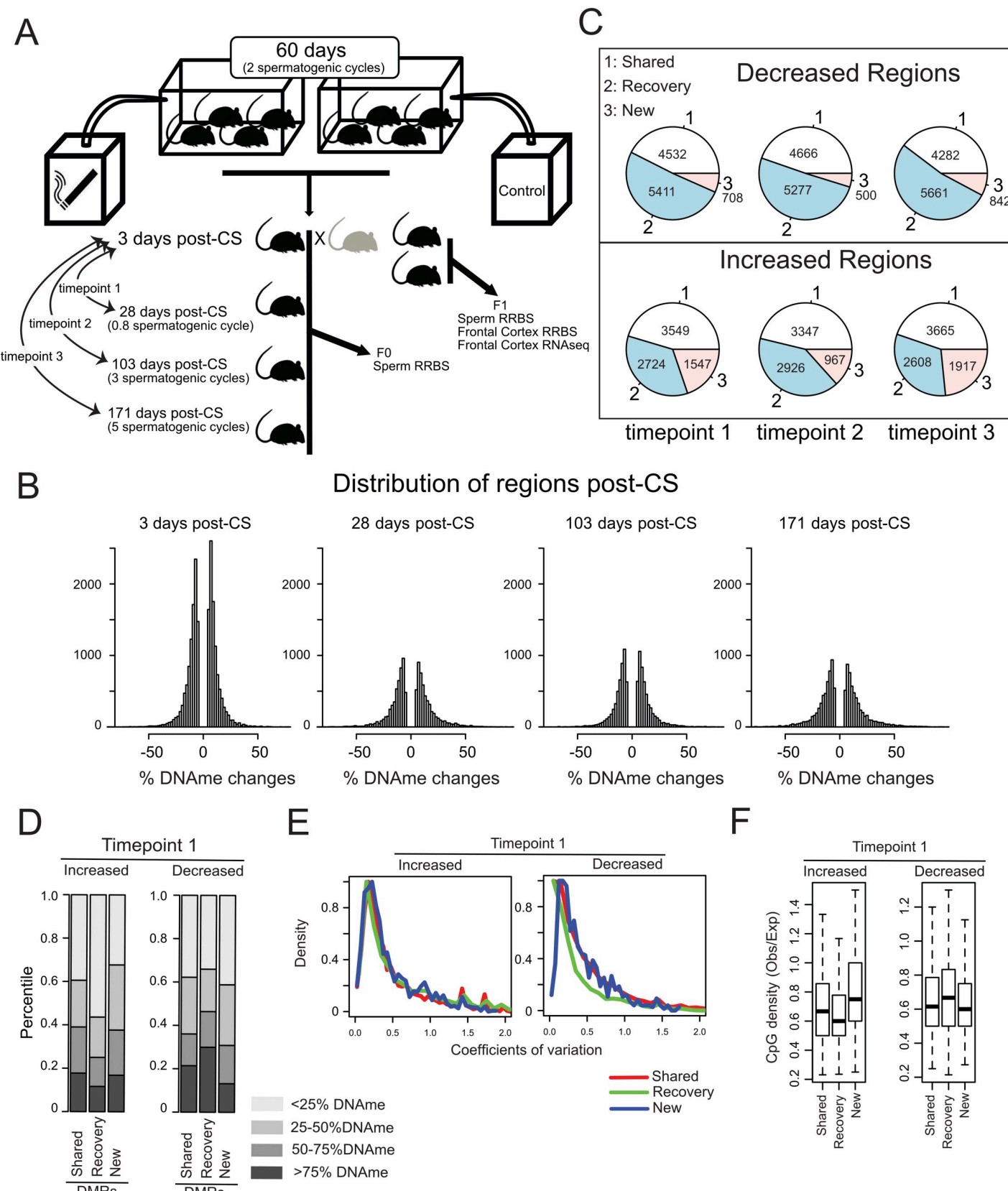

**Fig 1. Schematic of study design and regional DNA methylation changes and recovery. A)** Male mice were assigned to CS- exposed or non-exposed groups (n = 10–12 per group). The CS animals were exposed to the body mass-adjusted equivalent of 10–20 cigarettes per day, 5 days per week over a period of 60 days- corresponding to two complete cycles of spermatogenesis. CS- exposed and control mice were bred to unexposed females, and offspring were analyzed for phenotypic and molecular measures (see method for more details). **B)** Histograms describing the changes of DNAme in sperm after CS-exposure and recovery. **C)** The majority of DMRs observed prior to recovery were either maintained across the recovery period or returned to baseline levels, with only a small fraction of new DMRs emerging during the recovery period. Comparison is based on the 3 days post-CS group. **D)** Impact of the initial methylation status and direction of change on methylation recovery in the group analyzed 28 days after removal of CS. The hypomethylated DMRs (<25% DNAme) in which methylation increased and the hypermethylated DMRs (> 75% DNAme) in which methylation decreased with smoke exposure were more likely to recover. Regions of intermediate DNAme were less likely to recover. **E)** DMR recovery as a function of CpG density in the 28 days post-CS group. In every category of DMR (shared, recovery or new) variation diminished as CpG density of a region increased. The impact of CpG density on variation was particularly apparent for regions in which DNAme decreased as a result of CS exposure and later recovered to baseline. **F)** DMRs that displayed increased DNAme in CS-exposed animals and recovered within 28 days post-CS were generally regions of lower CpG density, while DMRs that lost methylation and subsequently recovered were generally at regions of higher CpG density.

based on their proximity (see methods). We then performed analyses similar to those performed for individual CpGs. Only regions with more than 3 CpGs were included, and all CpGs per regions were required to have a minimum of 8 reads in at least 4 biological replicates. As expected, changes in DNAme occurred at far fewer regions compared with individual CpGs (S1E Fig), and importantly, we found strong evidence for recovery of smoke-induced sperm DNAme changes following removal of the exposure. Indeed, the majority of DNAme regions that recovered returned to control levels within 28 days of removal of smoke exposure, and additional recovery was not observed following longer recovery periods (Fig 1B). In contrast with the individual CpG data, we found that far fewer new differentially methylated regions (DMRs) emerged during the recovery period (Fig 1C).

We then sought to understand epigenetic properties that might impact the ability of specific regions to recover following removal of the CS. For this analysis, we classified all DMRs into six classes, based on their dynamics during recovery (shared, recovered or new) and their direction of change (increase or decrease). We discovered that upon smoke exposure the hypomethylated regions where DNAme increased, and hypermethylated regions where DNAme decreased, were more likely to recover (Fig 1D and S2A Fig). Intermediate DNAme were less likely to recover across all groups, thus suggesting that regions of extreme hyper- and hypo-DNAme were less likely to change after CS exposure, and when changes did occur, these regions were more likely to recover. We then investigated whether normal variation in DNAme might contribute to the CS mediated DNAme changes. Indeed, DMRs where methylation changes were maintained tended to have higher overall variation and a higher degree of statistical significance (S2B Fig). We observed that recovered DMRs which initially decreased in DNAme level had lower variation compared with all other groups (Figs 1E and S2C). Moreover, DMRs that displayed increased DNAme in CS-exposed animals and subsequently recovered were generally regions of lower CpG density, while DMRs that lost DNAme and subsequently recovered were generally at regions of higher CpG density (Fig 1F and S2D Fig).

## Paternal CS exposure impacted DNAme and gene expression changes in F1 brains

We next sought to determine whether paternal CS exposure had any effect on offspring phenotype. Given that brain and nervous systems were previously reported to be sensitive to preconception paternal exposures [25–28], we investigated DNAme by RRBS in the prefrontal cortex of F1 mice derived from CS-exposed sires compared to the offspring of unexposed males (n = 8 per group). These experiments were performed exclusively on male mice in order to avoid confounding due to sex-related differences in DNA methylation and RNA expression. We found that paternal smoking altered DNAme patterns in over 28,000 regions in the prefrontal cortex of offspring (Fig 2A).

To further investigate the potential for phenotypic effects in offspring associated with paternal CS exposure, we performed RNA-seq of the F1 prefrontal cortex in the same animals as those assessed for DNAme. We identified 134 genes that were significantly differentially expressed ($|log2(fold change)|>1$ and p-value $< 0.05$) in F1 offspring sired by CS exposed males compared with unexposed males. To determine whether the observed gene expression changes were related to differential methylation change, we attempted to score gene expression changes at DMRs located within 1kb of transcription start sites (TSS). Unfortunately, this resulted in only 7 DMRs occurring within 1kb of any differentially expressed genes (p-value = 0.1, see methods). Thus, we were unable to conclude that DNAme changes were associated with gene expression changes, likely due to a lack of RRBS coverage at promoter regions.

Interestingly, paternal CS exposure caused a globally elevated variation in gene expression in the F1 brains (S2E Fig). This increased variation somewhat limited our ability to reliably identify differentially expressed genes. To explore potential functional consequences of altered gene expression, we ranked genes by change in gene expression and performed gene set enrichment analysis (GSEA) and gene ontology (GO) analysis. Using GSEA, we found that gene expression increased for genes associated with oxidative stress (Fig 2B and S1 Table). Using GO analysis, gene transcripts associated with gene classes including immune response and metabolism were over-represented, and transcripts associated with neuropeptide receptors and hormone activity were significantly under-represented in the offspring of CS-exposed males (Fig 2C).

## Paternal CS-induced effects are likely not transmitted beyond the first generation

To investigate whether the impact of CS exposure could be passed to subsequent generations, we compared the smoke-associated DNAme changes in F0 sperm with those of F1 sperm and found no correlation (r = -0.001; Fig 2D). As expected, we also found that paternal CS-associated DMRs in F1 brains and those in smoke-exposed F0 sperm had minimal overlap (hypergeometric p-value = 1; Fig 2E), indicating that F0 sperm DNAme changes do not directly impact DNAme levels in the F1 prefrontal cortex, consistent with the DNAme erasure that occurs during early mouse development. Accordingly, regions where DNAme changes occurred are not marked by chromatin features that are known to be maintained in mature sperm [29] (S3 Fig). While the DNAme effects described here can certainly be considered a marker of epigenetic impacts of CS, these results suggest that the effects of CS exposure likely do not persist beyond the first generation.

## CS-induced epigenetic changes were similar to those observed in Nrf2$^{-/-}$ mice

Based on our GSEA studies (Fig 2B), we hypothesized that impacts of CS exposure might be due in part to oxidative stress. This led us to examine F0 sperm DNAme changes in *Nrf2*$^{-/-}$ mice (compared to WT mice), as well as CS-exposed mice compared with unexposed *Nrf2*$^{-/-}$ mice. If oxidative stress played a role in the sperm DNAme changes, the changes observed in *Nrf2*$^{-/-}$ mice would mirror changes in CS-exposed WT mice, and impact on CS-exposed *Nrf2*$^{-/-}$ mice might be more pronounced. In agreement with our hypothesis, we found that CS-induced DNAme changes in WT sperm displayed significant similarity to those in *Nrf2*$^{-/-}$ sperm independent of CS exposure (Fig 3A, 3B and S4A Fig). Furthermore, CS exposure had no additional impact on *Nrf2*$^{-/-}$ sperm DNAme levels (Fig 3C), and association with CS exposure was maintained independent of the increased variation we observed in WT exposed mice (S4B Fig). Notably, these correlations were not detected when comparing our data to similar

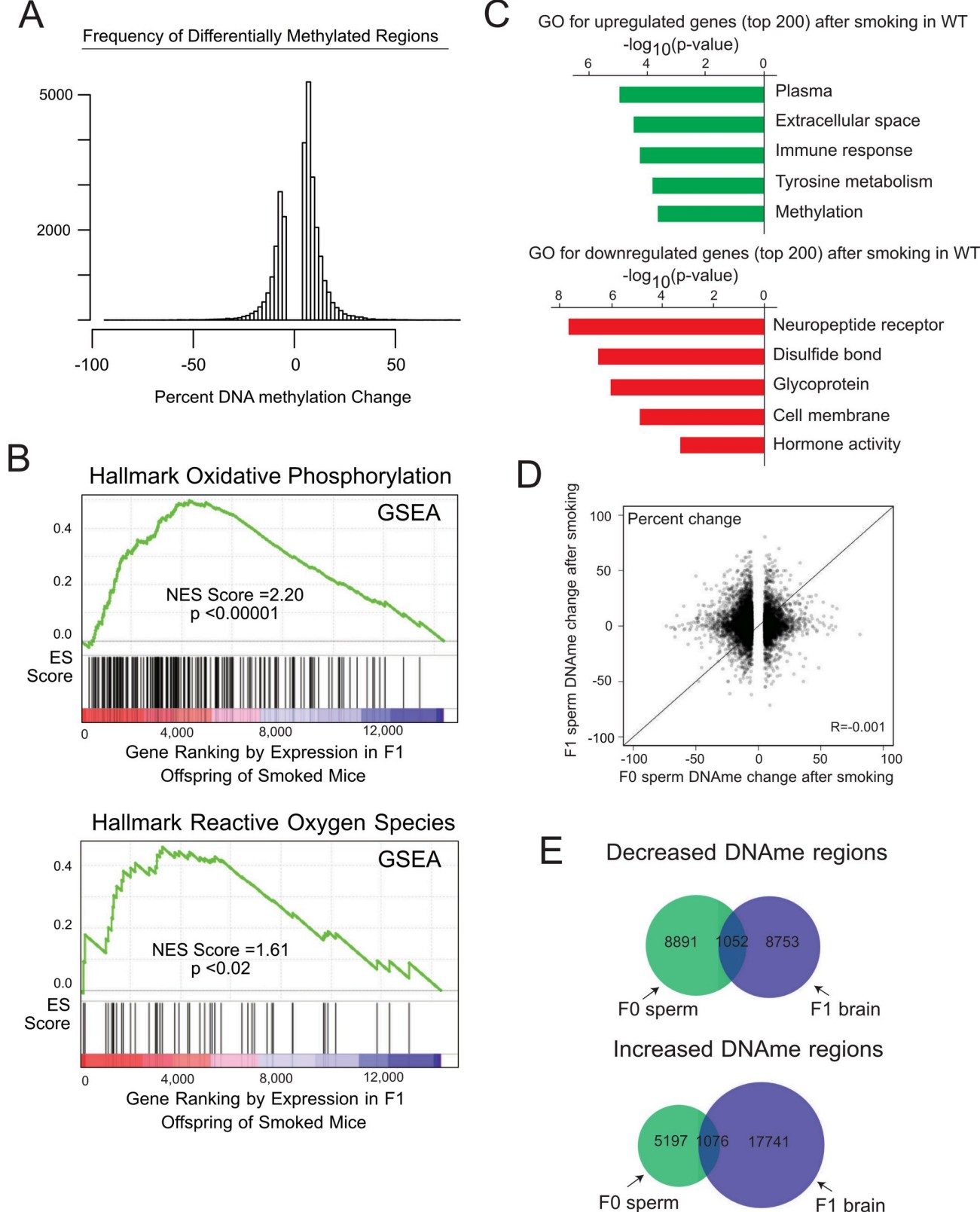

**Fig 2. Cigarette smoking leads to changes in gene expression in offspring brain, but these changes do not suggest a direct DNAme inheritance model. A)** Variation in F1 prefrontal cortex gene expression was significantly higher in CS-exposed animals compared with controls suggesting

stochastic dysregulation of gene expression in paternal CS-exposed offspring and offspring of mice with reduced antioxidant capacity. **B)** GSEA analysis revealed that oxidative phosphorylation pathway genes (top) and reactive oxygen species genes (bottom) are significantly upregulated in the brains of the F1 offspring of smoked mice. **C)** Gene ontology analysis of significantly upregulated and downregulated genes associated with paternal CS exposure indicated significant overrepresentation of several gene families. **D)** The DNAme changes observed in F0 sperm were not observed in the sperm of F1 offspring, suggesting the CS-associated effects likely do not confer risk beyond the first generation. **E)** Likewise, no significant overlap was observed in DMRs in F0 sperm compared with DMRs in F1 prefrontal cortex.

datasets evaluating the effect of other environmental factors (vinclozolin exposure and protein restricted diet) on sperm DNAme, suggesting independent mechanisms function during CS-exposure [30, 31] (S4C Fig).

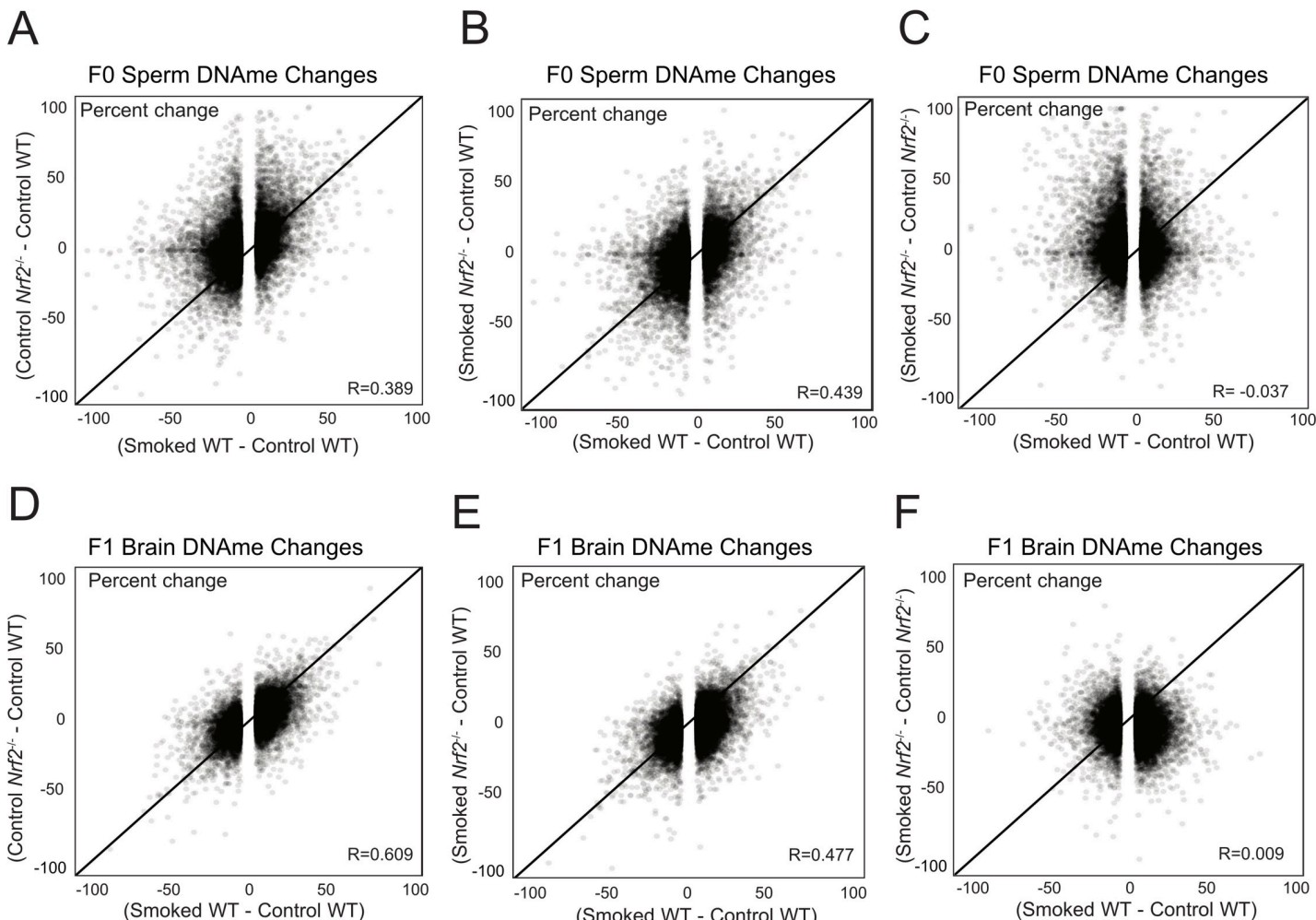

**Fig 3. Differential methylation in F0 sperm and F1 prefrontal cortex. A)** CS-induced DNAme changes in WT sperm displayed significant similarity to those in unexposed Nrf2$^{-/-}$ sperm. **B)** Similarly, sperm from CS-exposed *Nrf2*$^{-/-}$ mice displayed DNAme patterns similar to those of CS-exposed WT mice. **C)** However, the sperm DNAme changes in *Nrf2*$^{-/-}$ mice that were attributable to CS-exposure were not associated to the CS-associated changes in WT sperm, suggesting that genotype in *Nrf2*$^{-/-}$ mice is the primary driver of DNAme change in regions impacted by CS in WT mice. **D)** A highly significant correlation was observed in F1 brain DNAme changes induced by CS exposure in WT sires (x-axis) compared with DNAme changes associated with paternal *Nrf2* status, even in the absence of CS-exposure. **E)** A similar correlation in DNAme change was observed in the offspring of CS-exposed *Nrf2*$^{+/-}$ mice. **F)** The correlation disappeared when evaluating differential methylation in *Nrf2*$^{+/-}$ offspring based on CS exposure status compared with CS-associated DNAme changes in offspring sired by WT mice.

### The effects observed in CS-exposed offspring are similar to those in *Nrf2*^-/- offspring

Based on our prior results that CS-exposure impacts DNAme and gene expression patterns of offspring, we wondered whether these CS-induced F1 effects might be similar to impacts observed in offspring of *Nrf2*^-/- males. Indeed, the DNAme changes we observed in *Nrf2*^-/- F1 prefrontal cortex were highly similar to changes observed in the offspring of CS-exposed WT mice (Fig 3D and 3E, r = 0.609 and r = 0.477 respectively; and S2 Fig). In agreement with F0 sperm DNAme data, exposure to CS appeared to have no additional impact on brain DNAme beyond the *Nrf2*^-/- effect (Fig 3F). These observations strongly suggest that CS-induced DNAme changes that occur in F1 brains are largely mediated by paternal oxidative stress [32, 33].

We next sought to determine whether the CS-induced impact on gene expression patterns in F1 brain could also be associated with oxidative stress, as in F0 animals. Similar to our observations in the offspring of CS-exposed WT males, we found globally elevated variation in prefrontal cortex gene expression in the *Nrf2*^+/- offspring (S4D Fig). In addition, we observed a significant correlation in gene expression between offspring of CS-exposed WT mice and control *Nrf2*^-/- mice (Fig 4A) as well as between CS-exposed WT and *Nrf2*^+/- offspring (Fig 4B). Similar to our DNAme measurements, the effects were not further elevated in the offspring of smoke-exposed Nrf2^-/- mice (Fig 4C). Although the magnitude of gene expression change was modest, the offspring of *Nrf2*^-/- mice showed significant similarities with gene expression changes observed in the offspring of CS-exposed WT mice based on GO analysis (Fig 4D) and highly significant overlap in differentially expressed genes (Fig 4E). Additionally, GSEA analysis indicated gene sets commonly activated in response to inflammation and associated with cancer were elevated in both sets of F1 offspring (Fig 4F and S1 Table).

## Discussion

In this work, we studied the effect of CS exposure on sperm DNAme. We found many of the DNAme effects that recovered occurred at CpG-dense regions, which are enriched for genomic regulatory regions (such as promoters and enhancers). Thus, DNAme changes at these regions, which might otherwise impact gene regulation, appear to be relatively short-lived. Our observation that individual CpGs, and low CpG density regions did not recover to nearly the same degree as CpG dense regions may offer insight into the role of increased CpG density at promoters in maintaining fidelity of gene expression. It is possible that increased control and recovery of DNAme that we observed at CpG rich regions is the result of CpG-density acting to "buffer" against environmental insults. These data agree with previous reports that CS exposure significantly impacts DNAme patterns in whole blood, and CS-associated DNAme changes are largely corrected following smoking cessation in a time-dependent manner [34, 35]. We observed recovery after just twenty-eight days, which corresponds to just less than the period of a full spermatogenic cycle in mice of 30 days. In humans, a spermatogenic cycle is 67 days, and additional research is required to characterize the similarities and differences in the dynamics of sperm DNAme alterations between mice and men. Additionally, it is not yet known whether the corrections observed following removal of CS exposure ameliorate the affects observed in offspring, or whether those effects are driven by the regions that persist following CS removal, prompting further investigation.

We further investigated the impact of CS exposure in the oxidative stress-compromised *Nrf2*^-/- mice and found the CS-effects observed in WT animals were similar to those in *Nrf2*^-/- mice independent of CS exposure, suggesting elevated oxidative stress as a likely mechanism for CS-mediated sperm epigenetic changes. These findings were consistent with the

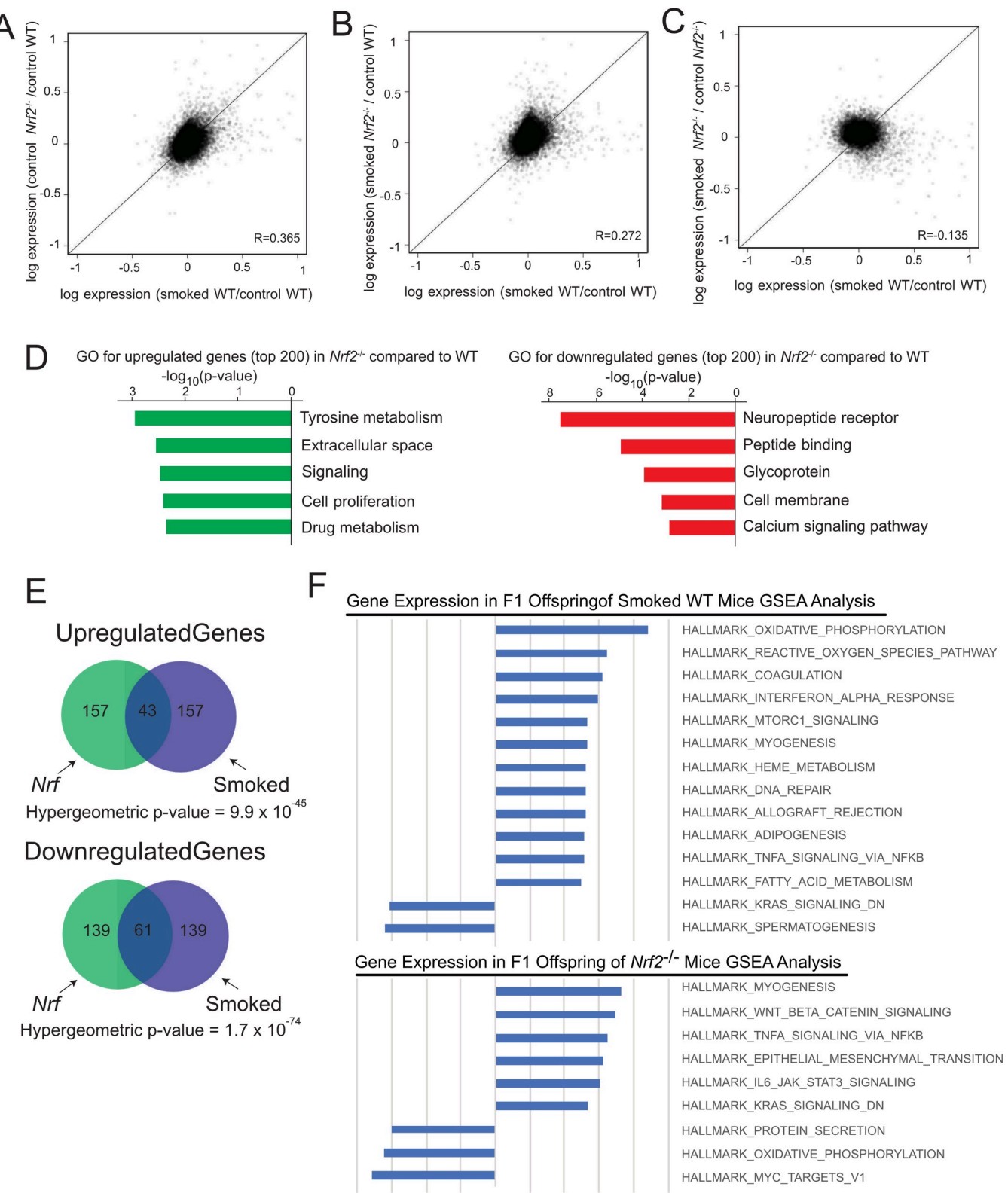

**Fig 4. Prefrontal cortex gene expression variation and correlation closely reflect the themes observed in F0 sperm. A)** A significant correlation in gene expression was observed between offspring of CS-exposed WT mice and control $Nrf2^{-/-}$ mice. **B)** Likewise, the correlation was observed between CS-exposed WT and $Nrf2^{-/-}$ offspring. **C)** However, there was no correlation in gene expression changes when comparing offspring of CS-exposed and unexposed $Nrf2^{-/-}$

mice. **D)** GO-terms for genes whose expression change was associated with *Nrf2*$^{-/-}$ genotype were similar to those associated with paternal smoking status in WT offspring. **E)** We found a highly significant overlap in differentially expressed genes associated with paternal smoking status in WT animals compared with offspring of unexposed *Nrf2*$^{-/-}$ mice. **F)** Pathways that are up- or down-regulated in the offspring brain of smoked WT mice (top) or *Nrf2*$^{-/-}$ mice (bottom), based on GSEA analysis.

observations of offspring prefrontal cortex DNAme and gene expression changes. The increased variation in prefrontal cortex gene expression, both in offspring of CS-exposed WT mice as well as unexposed *Nrf2*$^{-/-}$ mice, is noteworthy. This may be related to the increased variation in sperm DNA methylation in the sires, suggestive of somewhat stochastic impacts on sperm epigenetics. Interestingly, recent studies of two epigenetic modifiers, *Dnmt3a* and *Trim28*, have shown that differential expression of these factors can result in increased variability of gene expression patterns [36, 37]. Notably, *Trim28* and *Dnmt3a* expression were not markedly different between groups in the current study. The exploration of the role of these and related factors in the context of oxidative stress warrants further investigation. While CS-exposure may represent an extreme example of an environmental insult that induces oxidative stress, the list of environmentally relevant exposures that increase oxidative stress is extensive. The results presented here indicate that all such exposures could potentially impact the epigenetic status of the paternal germline and thus offspring phenotype. In addition, our work provides strong evidence that *Nrf2*$^{-/-}$ mice can serve as an important animal model to study CS-exposure induced effects. Further studies may focus on investigating the molecular pathways underlying NRF-mediated DNAme alterations.

By comparing the DNAme and gene expression patterns in the prefrontal cortex between offspring of male mice exposed to CS with those not exposed, we found strong evidence for an impact of paternal smoking on offspring phenotype. We suggest that paternal exposure to inducers of oxidative stress might contribute to behavioral or developmental impacts in offspring. This is supported by GSEA results that showed significant up-regulation of a number of cancer- and early development-associated pathways (e.g. INFα, TNFα, Jak-STAT and Wnt/beta catenin; Fig 4F). Notably however, our data showed little overlap of DNAme changes in the F1 prefrontal cortex with DNAme changes in the F0 sperm. While this was somewhat expected, as DNAme state undergoes dramatic reprogramming during early development and neuronal differentiation, it is important to highlight—direct mitotic inheritance of DNAme state is unlikely to mechanistically contribute to the effects we observed in F1 mice. This finding is consistent with previous studies investigating the transmission of sperm DNA methylation changes [38, 39]. We propose that DNAme may be more accurately described as a marker of epigenetic inheritance and not a mechanistic driver in transmitting environmental impacts to subsequent generations. Additional studies are necessary to confirm this and to distinguish other epigenetic features as markers or drivers, including chromatin status and small RNAs, which could play a significant role in inheritance. In particular, a growing body of evidence suggests that transfer RNA-derived small RNAs (tsRNAs) are an important paternal epigenetic factor that may underlie sperm-mediated epigenetic inheritance and represent an important area of investigation in the context of the data presented here [40–43]. Interestingly tRNA cleavage is induced by oxidative stress [44], and recent studies have demonstrated that dietary changes can have acute impacts on human sperm tsRNAs [45, 46]. One interesting possibility is that these dietary impacts occur through an oxidative stress mediated mechanism. It is important to emphasize however that we did not identify CS-associated DNAme changes in the sperm of the F1 generation, providing reassuring evidence that the changes observed in F1 animals likely would not persist in the F2 generation. However, direct studies to evaluate the potential for transgenerational impacts of CS exposure, and possible impacts in humans, are warranted.

We have presented evidence for the impact of preconception paternal CS exposure on sperm epigenetic profiles and offspring phenotype and highlight compelling similarities in offspring of *Nrf2*-null animals independent of CS exposure. We propose that these similar effects are related to the underlying mechanism by which CS mediates its effects on male gametes–through elevated oxidative stress or compromised capacity to maintain redox balance. Future studies should evaluate oxidative stress and redox capacity directly to better characterize the role of these factors in driving the observed effects and to more specifically define the mechanism(s) by which these types of changes are induced. In addition, since males were directly mated with females (rather than through IVF), it is formally possible that some of the changes observed in offspring were due to factors related to maternal investment based on olfactory or other cues indicating a poorer quality mate [47]. In this regard, further studies are necessary to validate the current findings and to more fully explore the mechanisms underlying our observations.

Importantly, the results of our study must be considered within the context of the greater scientific community, and thus, we caution against over interpretation. Studies in mouse often do not translate beyond, and therefore, further epidemiological or clinical studies are required to measure the heritable impacts of oxidative stressors in humans. Additionally, studies similar to ours, carried out by independent research groups, using similarly rigorous DNA methylation analysis methods and several biological replicates, are necessary to validate our findings in mice. These types of replication studies are particularly important in the epigenetics community, as several reports have questioned how frequently epigenetic inheritance occurs in mammals and humans [48–50]. It will also be important for future studies to determine the degree to which widespread oxidative stressors cause epigenetic impacts to individuals and on subsequent generations. If indeed our results are directly translatable to humans, then it will be critical for researchers to more fully explore the mechanisms underlying the observed epigenetic changes.

Our study has important implications in characterizing the potential mechanisms that underlie the elevated health risks observed in offspring of men who smoked prior to conception. The findings reported here may also be useful in understanding the risks of other environmental exposures that induce oxidative stress such as air pollution and some chemical exposures. Paternal preconception exposures to a variety of pharmacologic agents and pollutants, including nicotine, THC, morphine, and benzo[a]pyrene affect offspring phenotype, and often confer neurobehavioral consequences [25–28, 51–54]. In some cases, the impacts are transmitted across multiple generations [52, 53]. In this study, the absence of correlation between DMRs in F0 sperm and F1 sperm suggests that the affects we observed are likely not transmitted beyond the first generation, but we cannot rule out an alternate mechanism for transgenerational inheritance, including noncoding RNAs and chromatin features. While additional studies are necessary to fully characterize the long-term impacts of CS exposure, the current study significantly expands our understanding of paternal CS exposure impacts on offspring, and identifies a plausible mechanism underlying CS-induced epigenetic changes in sperm.

## Materials and methods

### Animals

**Ethics statement.**   All animal experiments were performed under protocols that were approved by the University of Utah Institutional Animal Care and Use Committee (protocol # 14–11006). All animals were obtained from Jackson Laboratories (Bar Harbor, ME)

**Experimental design.**   Six to seven-week-old mice were assigned to one of two groups: CS-exposed mice and non-exposed control mice. Following 60 days of CS exposure, mice were

bred to unexposed CAST/EiJ female mice. Groups of animals were euthanized and tissues collected 3, 28, 103, and 171 days after removal from CS exposure. Sperm DNA methylation analysis was performed by RRBS on F0 exposed and control animals. Offspring derived from exposed and control males were euthanized at 14–17 weeks of age, and tissues were collected. DNA methylation analysis was performed on sperm and prefrontal cortex to investigate the impact of paternal smoking status on methylation patterns in offspring. In addition, RNAseq was performed on prefrontal cortex tissue to investigate the association between paternal smoking and gene expression.

**Smoke exposure.**   All CS-exposed and control mice were age matched and smoking was initiated between 6 and 7 weeks of age. Mice were exposed to CS using a Teague Model TE-10 (Teague Enterprises, Woodland, CA) smoking machine, which produces a combination of side-stream and mainstream CS. A pump on the machine "puffs" each 3R4F University of Kentucky research cigarette for 2 seconds for a total of 9 puffs before ejection. The 2.5-hour daily exposure occurred for 5 consecutive days per week over a period of 60 days. The smoking chamber atmosphere was periodically sampled to confirm total particulate matter concentrations of approximately 150 mg/m$^3$, the human equivalent of smoking approximately 10–20 cigarettes per day [55].

**Smoking and recovery experiments.**   To characterize the impact of smoke exposure on the sperm DNA methylome, and the capacity for smoke-induced sperm DNAme alterations to recover following removal of the insult, we exposed 40 C57BL/6J (Jackson Labs Stock # 000664) to cigarette smoke for comparison against 10 age-matched, non-smoked controls. Ten CS-exposed mice and the 10 non-exposed controls were euthanized and tissues collected within three days of the CS exposure period. Subsequent "recovery" groups of 10 CS-exposed animals were euthanized 28, 103 and 171 days after the exposure period (corresponding to approximately 0.8, 3 and 5 spermatogenic cycles). In addition to experiments with WT animals, ten age-matched *Nrf2*$^{-/-}$ mice on a C57BL/6J genetic background (Jackson Labs Stock # 017009) were exposed to the same doses of CS for the same time period, and ten age-matched unexposed *Nrf2*$^{-/-}$ mice were utilized as controls.

**Offspring transmission experiments.**   Founder mice for heritability experiments included WT C57BL/6J mice (Jackson Labs Stock # 000664) that were exposed and not exposed to CS (n = 10–12 per group). Approximately one week after the exposure period, exposed and control males were introduced to 6-week old CAST/EiJ female mice (Jackson Labs Stock # 000928), and pairs were kept together until F1 litters were born, or for 7 weeks without conceiving, whichever came first. The motivation for outcrossing males to CAST/EiJ females was to leverage polymorphic alleles to enable attributing reads to a specific parent, however due to the large average spacing of informative SNPs in the CAST strain and the short sequencing reads inherent in Illumina sequencing we were unable to classify the large majority of reads based on parent-of-origin. We therefore analyzed the data without regard to parent-of-origin. F1 litters were weaned at approximately 21 days of age, and pups were regularly weighed until they were euthanized. F1 animals were euthanized at 14–17 weeks of age, and heart, lung, liver, kidney, brain, testis and epididymal sperm were collected for molecular studies.

**Animal phenotyping.**   Following epididymal sperm extraction, sperm count and motility were assessed in CS-exposed and control F0 animals as well as F1 offspring. In addition, time to conception and litter size were compared between F0 groups. F1 offspring were evaluated for growth trajectory. For statistical analysis of growth trajectories between groups, animal weights were plotted against age for all pups within a group (C57BL/6J or *Nrf2*$^{-/-}$). Models to fit the data were tested, and a logarithmic model generally yielded the highest r$^2$. Theoretical weights were calculated for each weight event based on the model generated, and differences

between theoretical and actual weight were calculated. A mean of average differences within an individual across weight events was calculated for each animal, and unpaired student's t-test was used to compare these differences between smoked and non-smoked animals within each group. Differences in animal weights and sperm parameters were evaluated using two-tailed Student's t-test, and two-tailed Fisher's Exact tests were used to evaluate weekly differences in conception between groups. $P < 0.05$ was considered significant.

## Molecular analyses

**Sperm collection and DNA extraction.** Sperm was collected from the cauda epididymis and vas deferens immediately after euthanasia by scoring the tissue along the length of the tubules with a 28-G needle and gently pressing the tissue to expel the sperm mass. Tissues were then placed in a center-well dish in equilibrated Quinn's medium (CooperSurgical, Trumbull, CT) supplemented with FBS in a humidified $CO_2$ incubator for one hour. Following the swim out period, sperm concentration and motility were assessed on a Makler chamber and sperm were snap frozen in liquid nitrogen. Samples were subsequently thawed and subjected to a stringent somatic cell lysis protocol to ensure a pure population of sperm. Briefly, samples were passed through a 40 μM filter to remove cell and tissue clumps followed by two 14 ml washes with ddH$_2$O and incubation for at least 60 minutes in somatic cell lysis buffer (0.1% SDS, 0.5% Triton X in ddH$_2$O) at 4º C. Following somatic cell lysis and visual confirmation of the absence of contaminating cells, sperm DNA was extracted using the Qiagen AllPrep Universal kit (Hilden, Germany). Samples in cell lysis buffer were passed through a 28-gauge syringe multiple times to disrupt sperm membranes and liberate nucleic acids prior to extraction.

**Prefrontal cortex dissection and nucleic acid extraction.** Following euthanasia of F1 males (n = 8 per group), left brain hemispheres were dissected and placed in PreAnalytiX Pax-Gene (Hombrechtikon, Switzerland) tissue stabilizer and after 24 hours, fixed in PaxGene fixative and stored at -80º C. Samples were subsequently thawed and prefrontal cortex dissected under a stereo microscope according to the method described by Chiu et al. [56]. Tissue was then disrupted using a microcentrifuge pestle, and RNA and DNA were extracted using the Qiagen AllPrep Universal kit according to manufacturer's protocols.

**RRBS library construction.** Following DNA extraction, Bioo Scientific NEXTflex Bisulfite Library Prep Kit for Illumina Sequencing (PerkinElmer, Austin, TX) was used for library preparation. To maximize coverage, we employed two separate restriction digests with MspI and TaqαI. Following digestion, products were pooled, and Klenow Fragment was utilized to create 3'A overhangs. DNA was subsequently purified with Zymo DNA Clean and Concentrate Columns (Irvine, CA) followed by ligation of Methylated Illumina PE Adapters and Ampure purification with SPRI beads. Purified products were Sodium Bisulfite Converted using ZymoResearch EZ DNA Methylation Gold Kit, and libraries were amplified over 20 cycles using Platinum Taq DNA polymerase (ThermoFisher, Waltham, MA), followed by a final Ampure purification (Beckman Coulter, Indianapolis, IN) and confirmation of library size range on a 2% agarose gel. DNA was submitted to the Huntsman Cancer Institute High Throughput genomic core for sequencing on a Hi-Seq 2500 (Illumina, San Diego, CA) using 50 cycle-single read chemistry. Four to six samples were sequenced per lane for a minimum of 35-million reads per sample.

**RNAseq library construction.** RNA extracted from F1 frontal cortices (n = 8 per group) was subjected to Illumina TruSeq Stranded RNA kit with Ribo-Zero Gold library preparation and subsequently sequenced on a Hi-Seq 2500 using 50 Cycle-Single Read Sequencing v4. Eight samples were sequenced per lane for a minimum of 25-million reads per sample.

**Bioinformatics analyses.** For genome wide DNA methylation analysis, sequence data from RRBS libraries was aligned to the mouse mm10 genome using the Bismark pipeline with special attention to RRBS specific issues, as noted in the Bismark User Guide and the Bismark RRBS Guide. Only CpGs where read coverage was greater than 8 for at least 4 biological replicates were considered "scoreable" for downstream analysis. Only CpGs with more than 5% change in methylation relative to control samples were classified as differentially methylated. When considering DMRs, only regions greater than 50 base-pairs in length with 3 or more scorable CpGs were analyzed. Then, one third of the CpGs within each analyzed region needed to be differentially methylated in order for a given region to be under consideration as a DMR. Finally, qualifying regions were classified as bona fide DMRs if there was more than 5% change in methylation relative to control samples. For genome wide gene expression analysis, sequencing data from RNASeq libraries was aligned using Novoalign. Aligned splice junction were converted to genomic coordinates and low quality and non-unique reads were further parsed using SamTranscriptomeParser (USeq; v8.8.8) under default settings. Stranded differential expression analysis was calculated with the USeq program DefinedRegionDifferential-Seq, which utilizes DESeq2 and the reference mm10. Normalized read count tables were then analyzed in R, along with all DNA methylation data. We identified 28,707 regions that display >5% DNAme change in the smoked F1 brain vs control and 134 genes that display differential gene expression between smoked F1 brain vs control ($|Log2FC| > 1$ and p-value $< 0.05$). We performed intersection analysis by counting the number of DMRs that overlap with gene promoters (regions within 1kb of TSS). To calculate the p-value, we utilized the BedTool ShuffleBed function, and generated 134 random TSS regions, and intersected these with DMRs. This permutation was re-iterated 1000 times, allowing us to calculate the p-value for enrichment. Integration and parsing of bed files or tables was performed in R. Generation of all figures and statistical analyses was accomplished using standard methods in R, with the exception of aggregate histone modification profiles, which were generated using Deeptools. Gene ontology analysis was performed using DAVID Functional Annotation Bioinformatics Resources. GSEA was performed on normalized gene expression read count tables using software development at the Broad Institute (http://software.broadinstitute.org/gsea/index.jsp).

## Supporting information

**S1 Fig. Descriptive statistics of F0 animals and F1 animals and diminished recovery at individual CpGs.** A) F0 animals- Both WT and *Nrf2*$^{-/-}$ mice that were exposed to CS weighed significantly less than non-exposed control animals. Sperm concentration and motility were not significantly impacted by CS exposure, sperm concentration was lower in *Nrf2*$^{-/-}$ than WT males, and conception was significantly delayed in CS-exposed *Nrf2*$^{-/-}$ mice compared with unexposed *Nrf2*$^{-/-}$ mice. B) F1 animals- Neither growth trajectories nor sperm parameters were different in F1 animals based on paternal CS exposure. Abbreviations: CN-control *Nrf2*$^{-/-}$, SN-smoke-exposed *Nrf2*$^{-/-}$, CWT-control wild type animals, SWT-smoke-exposed wild type animals. C) Quantitative data indicate that the number of differentially methylated loci in CS-exposed mice compared with age matched controls does not diminish following a recovery period of up to 171 days. D) When considering only loci that lost methylation in the CS-exposed group, about one third of differentially methylated loci persisted for the entire recovery period (white), one third returned to baseline levels (blue) and one third emerged as differentially methylated following a recovery period (pink). Contrastingly, for loci that gained methylation as a result of CS exposure, a smaller fraction of differentially methylated CpGs persisted or recovered while nearly half of differentially methylated loci emerged during the recovery period. E) As expected, the number of differentially methylated CpGs and the

magnitude of change in DNAme was much higher than the changes observed regionally.
(TIF)

**S2 Fig. Summary of the recovery, persistence or emergence of differentially methylated loci (A and B) and properties of recovery regions (C, D and E)** A) Impact of the initial methylation status and direction of change on methylation recovery. The hypomethylated DMRs (<25% DNAme) in which methylation increased and the hypermethylated DMRs (> 75% DNAme) in which methylation decreased with smoke exposure were more likely to recover. Regions displaying methylation between 25% and 75% Regions of intermediate DNAme were less likely to recover across both groups. Both comparisons were to the 3 days post-CS group. Timepoint 2 = 103-day recovery group, and timepoint 3 = 171-day recovery group. Both comparisons were to the 3-day recovery group. Dynamics were similar to those observed at 28 days post CS (see Fig 1D and 1F). B) DMRs where methylation changes were maintained across the recovery period tended to display higher overall variation and a higher degree of statistical significance compared with regions that recovered. C) In every category of DMR (shared, recovery or new) variation diminished as CpG density of a region increased. The impact of CpG density on variation was particularly apparent for regions in which DNAme decreased as a result of CS exposure and later recovered to baseline. D) DMRs that displayed increased DNAme in CS-exposed animals and later recovered were generally regions of lower CpG density, while DMRs that gained methylation and subsequently recovered were generally at regions of higher CpG density. E) Variation in F1 prefrontal cortex gene expression was significantly higher in CS-exposed animals compared with controls suggesting stochastic dysregulation of gene expression in paternal CS-exposed offspring and offspring of mice with reduced antioxidant capacity.
(TIF)

**S3 Fig. Chromatin properties at F0 sperm DMRs and variation in F1 prefrontal cortex gene expression.** A) Chromatin accessibility was not predictive of the propensity for DMRs to recover or be maintained after removal of CS exposure. B-C) Localization of B) H3K4me3 and C) H3K27me3 was likewise not associated with DMR recovery or maintenance.
(TIF)

**S4 Fig. The effects observed in *Nrf2*<sup>-/-</sup> offspring are similar to those in CS-exposed offspring.** A) Heatmap illustrating the significant similarity between CS-associated sperm DMRs identified in WT mice and DMRs associated with the $Nrf2^{-/-}$ genotype, apparently independent of CS-exposure status. B) Association with CS exposure was maintained independent of the increased variation we observed in WT exposed mice C) A high correlation in DMRs was observed between CS-exposed WT mice and $Nrf2^{-/-}$ whether or not they were exposed to CS. No correlation was observed between DMRs identified in the current study compared with previously published DMRs associated with vinclozolin exposure (VD2) and protein restricted diet (PR). D) Variation in F1 prefrontal cortex gene expression was significantly elevated in $Nrf2^{-/-}$ controls compared with WT controls suggesting stochastic dysregulation of gene expression in offspring of mice with reduced antioxidant capacity.
(TIF)

**S1 Table. Gene Set Enrichment Analysis table.** Gene expression changes in smoked and Nrf2 mutant mice were analyzed using Gene Set Enrichment Analysis to determine if gene sets from the Hallmarks database were enriched. Enrichment scores (ES and NES) and statistical significance (p-values) are indicated for all measurements.
(XLS)

## Acknowledgments

We thank Dr. Kristin Murphy for thoughtful comments and critique during preparation of the manuscript.

## Author Contributions

**Conceptualization:** Timothy G. Jenkins, John R. Hoidal, Douglas T. Carrell, Bradley R. Cairns, Kenneth I. Aston.

**Data curation:** Patrick J. Murphy, Jingtao Guo.

**Formal analysis:** Patrick J. Murphy, Jingtao Guo, David F. Alonso, Kenneth I. Aston.

**Funding acquisition:** Timothy G. Jenkins, John R. Hoidal, Douglas T. Carrell, Bradley R. Cairns, Kenneth I. Aston.

**Methodology:** Patrick J. Murphy, Jingtao Guo, Timothy G. Jenkins, Emma R. James, Thomas Huecksteadt, Dallin S. Broberg, Kenneth I. Aston.

**Supervision:** Douglas T. Carrell, Bradley R. Cairns, Kenneth I. Aston.

**Visualization:** Patrick J. Murphy, Jingtao Guo.

**Writing – original draft:** Patrick J. Murphy, Jingtao Guo, Kenneth I. Aston.

**Writing – review & editing:** Timothy G. Jenkins, Emma R. James, John R. Hoidal, Thomas Huecksteadt, Dallin S. Broberg, James M. Hotaling, David F. Alonso, Douglas T. Carrell, Bradley R. Cairns.

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
