## [Decision Letter · Decision Letter 0]

3 Jan 2020

Dear Dr Murphy,

Thank you very much for submitting your Research Article entitled 'Oxidative stress underlies heritable impacts of paternal cigarette smoke exposure' to PLOS Genetics. Your manuscript was fully evaluated at the editorial level and by independent peer reviewers. The reviewers appreciated the attention to an important problem, but raised some substantial concerns about the current manuscript. Based on the reviews, we will not be able to accept this version of the manuscript, but we would be willing to review again a much-revised version. We cannot, of course, promise publication at that time.

Should you decide to revise the manuscript for further consideration here, your revisions should address the specific points made by each reviewer, including a detailed list of your responses to the review comments and a description of the changes you have made in the manuscript. We feel that there are issues raised by multiple reviewers that need to be addressed, in some cases by writing but in others by experiments. Specifically, the reviewers raised questions about the statistical methods (and also paucity of information on the analyses). They also wanted to see replication and validation by an independent method for interesting/high confidence loci. It is also strongly felt that these data should be compared with human data to determine if there are any loci in common that are susceptible to smoking (for example, see Jenkins et al, Andrology). Reviewers also would like to see a better correlation between the methylation and expression data in F1 brains and finally, a much more rigorous discussion of the results.

If you decide to revise the manuscript for further consideration at PLOS Genetics, please aim to resubmit within the next 60 days, unless it will take extra time to address the concerns of the reviewers, in which case we would appreciate an expected resubmission date by email to plosgenetics@plos.org.

[LINK]

We are sorry that we cannot be more positive about your manuscript at this stage. Please do not hesitate to contact us if you have any concerns or questions.

Yours sincerely,

Marisa S Bartolomei

Associate Editor

PLOS Genetics

Wolf Reik

Section Editor: Epigenetics

PLOS Genetics

Reviewer's Responses to Questions

**Comments to the Authors:**

Reviewer #1: This manuscript adds to the growing body of evidence that environmental effects in one generation can impact phenotypic and molecular outcomes in the next. Loosely speaking, this falls under the umbrella term of 'Lamarckian effects' to an extent. In this manuscript, the authors have used paternal cigarette smoking exposure as their environmental insult, and complement this with a genetic model Nrf2-/-. Overall, I think some of the results in the manuscript are certainly intriguing but I have several key concerns that need to be addressed prior to acceptance.

1. It doesn't seem as if the methylation analyses have been analysed in a rigorous manner. For example, considering the initial sperm methylation differences in sperm (Figure 1), are any of the lists of sites significant at the genome-wide corrected level. I appreciate that P values such as 0.05 do not demarcate truth from falsehood, but it does give the reader some sense of what to make of the data. In this case it would be especially important as the authors do not present an independent replication. In fact, I can't find any description of the stats in the Methods either.

2. In Figure 3, the authors say that CS induced meth changes in F0 sperm are very similar to those found in the NRF F0 sperm. But they only present heat maps. This needs to be redone as scatter plots just like they do for F1 mice in Figures 3B-D. On that note, these latter three analyses (B, C and D) are the most convincing and interesting in this manuscript.

3. If I am not mistaken, there are studies showing the impact of smoking on sperm in humans. Can the authors perform some comparisons using the published data? If there are similarities it would certainly strengthen the case.

4. In F1 brains, was there any correlation between methylation and expression?

5. I think in the discussion they should list the variety of shortcomings of their study. This in no way weakens their message but rather provides a more balanced view. For example, the bar now in this field is it perform IVF using sperm isolated from exposed F0 males so that the F0 males and females never come into physical contact (and eliminates other effects such as those potentially mediated by olfactory systems).

Reviewer #2: Murphy and colleagues provide a well-written manuscript "Oxidative stress underlies heritable impacts of paternal cigarette smoke exposure." They describe molecular phenotypes of paternal pre-conception cigarette smoke (CS) in the mouse. They find that paternal CS elicits paternal sperm DNA methylation changes as well as DNA methylation and gene expression alterations in offspring brain tissue. The functional categories of molecular changes in F0 sperm and F1 brain are enriched for oxidative stress pathways leading the authors to conclude that the influence of the sire's CS exposure is drive by oxidative stress. To support this claim, they compare the methylation and expression signatures of cigarette smoking with that of mice deficient in Nrf2 - a master regulator of endogenous antioxidant processes. They found positive correlations which they interpret as supporting their hypothesis. These experiments and analyses are important and have yielded useful information, but I have a number of concerns with the interpretation of the data, and the conclusions that are drawn, as described below.

Major Issues:

The conclusion that the paternal CS exposure "underlies" offspring molecular profiles is not sufficiently supported by the design and analyses. Throughout the manuscript, the authors use the language that the Nrf2 knockout mouse data "largely" recapitulate the effects seen in CS exposed sperm and offspring tissue. Largely implies a preponderance of signal that is shared. I agree that there is compelling overlap, but not that the it is recapitulated largely. Blocking the effect of paternal CS in F0 sperm or F1 somatic tissue with co-administration of antioxidants (e.g. tocopherol, or NAC) would support this mechanistic claim. Additionally, validating that there is indeed oxidative stress in the paternal germ cells or offspring brain (and that it is reflected similarly in the Nrf2-/- model) could provide additional direct support. In the absence of this, the claims have to be toned down throughout the manuscript, and the title (which currently implies causally supported results) and section headings of results sections would have to be modified considerably. For example, the section header on line 219 is particularly liberal with respect to the data.

The design of the study and the manuscript do not acknowledge paternal-independent effects on offspring molecular phenotypes. Dams may detect the exposure and invest differentially (e.g. PMID: 29514964) during gestation or lactation, which could contribute to offspring brain and sperm molecular signatures. In other words, the CS exposed males could represent poorer quality mates, or simply have altered olfactory signals, which leads the dams to invest less in the progeny. The only way(s) to account for this are to perform IVF and embryo transfer and/or perform cross-fostering to account for all maternal influences, and post-natal influences, respectively. I acknowledge that those experiments are often not feasible, so in such cases this possibility must be carefully acknowledged. Similarly, offspring sex and litter sex ratios have not been disclosed and should be. Alterations to sex ratios could alter molecular signatures in brain and sperm in F1.

Other Issues/Questions:

Was there any attempt to compare the mouse CS exposure DNAme and RNA-seq signatures to the previous human sperm publication data (e.g. the cited Jenkins et al., 2017 Andrology work)? If not, why?

RNA-seq data from prefrontal cortex samples are repeatedly listed as "neural" signals (abstract, line 35; pg 9 line 351), but the samples represent mixed cell types from bulk tissue, not just neurons.

Data from GSEA enrichment from ranked RNA-seq only include the two (Oxphos and ROS) (Fig 2E). These are certainly significant enrichment scores, but where do they rank amongst all MSigDB gene sets? Without the data on the other sets, your emphasis of those two could be seen as "cherry picking." A supplemental table listing all significant Gene sets, NES scores, and p-values (positive or negative enrichment) would provide appropriate transparency.

I do not feel that global gene expression variability should be interpreted as an effect or outcome. I recommend focusing on the enrichment and ontology analyses instead.

There are no validation experiments to validate DMRs (e.g. targeted bisulfite seq) or differentially expressed genes (e.g. qPCR) between treatment groups and/or genotypes. Accordingly, the molecular profiles have to be considered preliminary effects until such time they are validated.

The statement on page 5 in the second paragraph (line 183) stating that recovery influences suggest CpG density has something to do with buffering environmental influences is a bold statement. It is an interesting idea, but perhaps it is safer to say that DNAme across CpG regions may buffer environmental influences? You may need to clarify what you mean better, provide some sort of support, or move it to the discussion with appropriate level of language stating that is speculative.

Figure S3 panel D should have labels on the x-axis as well.

Reviewer #3: This is an overall well-performed study expanding our knowledge regarding the effect of cigarette smoking (CS) and sperm DNAme and its effect in the offspring. The author also reach a conclusion that the sperm and offspring DNAme changes induced by CS can by recapitulated by NRF2 deletion, which are known to increase oxidative stress. These data suggest that oxidative stress could underlies CS-induced sperm epigenetic changes and the effect in offspring. These are novel finding that contribute to the filed. However, I have some concerns and suggestions for the authors to consider for improvements.

1. First and foremost, the abstract should be reconstructed to be more informative on the major findings and to be more accurate on the results.

For example, the long sentence “This study used mouse models to evaluate: 1)…2)…3)…4)…” is not fully necessary and can be reduced or removed, the saved space could be used to describe more results.

The sentence “paternal smoking causes changes in neural DNAme and gene expression in offspring” is not accurate and potentially misleading, as the changes in F1 neural DNAme and gene expression in offspring, in fact, shows minimal overlapping with the paternal CS-associated DMRs in in smoke-exposed F0 sperm. This is an important piece of information and should be clearly stated in the abstract or it would be misleading.

The author also didn’t find overlapping between the DNAme between F0 and F1 sperm, which is consistent with the previous finding in other mammalian epigenetic inheritance models (Science 2014, PMID: 25011554; Genome Biol 2015, PMID: 25853433), and this result should be highlighted in the abstract. In fact, these results suggest DNAme may not be a direct vector in transmitting epigenetic phenotypes (it could be a marker representing changes in sperm epigenome but could be secondary effect), and his has triggered the emerging studies that focusing on sperm RNAs as the epigenetic information carriers in the transmission of phenotype, as extensively discussed recently (Nat Rev Genet 2016, PMID:27694809; Nat Rev Endocrinol 2019, PMID: 31235802). These should be discussed in the manuscript to make an informative paper.

2. In the results, the authors showed that “Interestingly, paternal CS exposure caused a globally elevated variation in gene expression in the F1 brains (Figure 2A)”, this is an interesting finding and may intricately related to epigenetic mechanisms. I suggest the authors to read recent publications on the case of Trim28 (Genome Biol 2010, PMID: 21092094; Cell 2016 2016, PMID: 26824653) which may shed new lights on the current study and should be at least discussed. The author may also check their DNAme and RNA-seq data regarding the status of Trim28.

Did the authors find any correlation between F1 brain DNAme and transcriptome changes? This point is not clearly stated. If the conclusion is negative, relate discussions should be provided. There could be many explanations.

3. The effect of NRF2 deletion on F0 sperm and F1 brain that mirrors the CS-exposure is striking, this could be due to oxidative stress and it raise many interesting questions. The effect could involve additional epigenetic changes in the sperm if the oxidative stress is the main effector. For example, oxidative stress is a strong inducer of tRNA cleavage and will generate more tsRNA and rRNA fragments (RNA 2018, PMID:18719243), and both tRNA and rRNA fragments (tsRNAs and rsRNAs) have been involved in sperm mediated intergenerational epigenetic inheritance of phenotypes (Science 2016, PMID:26721680; Science 2016, PMID:26721685; Nat Cell Biol 2018, PMID:29695786). These are interesting clues can be discussed in the paper.

**Have all data underlying the figures and results presented in the manuscript been provided?**

Reviewer #1: Yes

Reviewer #2: No: Complete enrichment score data from GSEA are not present.

Reviewer #3: Yes

PLOS authors have the option to publish the peer review history of their article (what does this mean?). If published, this will include your full peer review and any attached files.

Reviewer #1: No

Reviewer #2: Yes: Brandon Pearson

Reviewer #3: No

---

## [Decision Letter · Decision Letter 1]

16 Mar 2020

Dear Dr Murphy,

Thank you very much for submitting your Research Article entitled 'NRF2 loss recapitulates heritable impacts of paternal cigarette smoke exposure' to PLOS Genetics. Your manuscript was fully evaluated at the editorial level and by independent peer reviewers. The reviewers appreciated the attention to an important topic but identified some aspects of the manuscript that should be improved.

We therefore ask you to modify the manuscript according to the review recommendations before we can consider your manuscript for acceptance. Your revisions should address the specific points made by each reviewer. **Most importantly**, it is essential that you address the question of replication, as now raised by two reviewers. Also, please address the other suggestions. 

[LINK]

Yours sincerely,

Marisa S Bartolomei

Associate Editor

PLOS Genetics

Wolf Reik

Section Editor: Epigenetics

PLOS Genetics

Reviewer's Responses to Questions

**Comments to the Authors:**

Reviewer #1: The manuscript is certainly improved. However, the key shortcoming of replication (which looks like another referee raised) has not been addressed.

Reviewer #2: The authors made satisfactory edits in response to the issues I raised.

Reviewer #3: I'm overall satisfied with the revision and recommend acceptance for publication in Plos Genet.

I have only one more suggestions for consideration:

Since cigarette smoke is a human activity and the authors have discussed the potential link with sperm tsRNAs, it would be good to include the recent papers about the acute dietary effect on human sperm tsRNAs and the related discuss on how this could be regulated by oxidative stress (Plos Biol 2019, PMID 31877125; Nat Rev Endocrinol 2020, PMID:32066893)

**Have all data underlying the figures and results presented in the manuscript been provided?**

Reviewer #1: Yes

Reviewer #2: Yes

Reviewer #3: Yes

PLOS authors have the option to publish the peer review history of their article (what does this mean?). If published, this will include your full peer review and any attached files.

Reviewer #1: No

Reviewer #2: No

Reviewer #3: No

---

## [Editor Report · Decision Letter 2]

3 Apr 2020

Dear Dr Murphy,

We are pleased to inform you that your manuscript entitled "NRF2 loss recapitulates heritable impacts of paternal cigarette smoke exposure" has been editorially accepted for publication in PLOS Genetics. Congratulations!

Yours sincerely,

Marisa S Bartolomei

Associate Editor

PLOS Genetics

Wolf Reik

Section Editor: Epigenetics

PLOS Genetics

Comments from the reviewers (if applicable):

**Data Deposition**

http://datadryad.org/submit?journalID=pgenetics&manu=PGENETICS-D-19-01866R2

**Press Queries**

---

## [Editor Report · Acceptance letter]

1 Jun 2020

PGENETICS-D-19-01866R2 

NRF2 loss recapitulates heritable impacts of paternal cigarette smoke exposure 

Dear Dr Murphy, 

We are pleased to inform you that your manuscript entitled "NRF2 loss recapitulates heritable impacts of paternal cigarette smoke exposure" has been formally accepted for publication in PLOS Genetics! Your manuscript is now with our production department and you will be notified of the publication date in due course.

With kind regards,

Kaitlin Butler

PLOS Genetics

On behalf of:
